# LVBench: An Extreme Long Video Understanding Benchmark

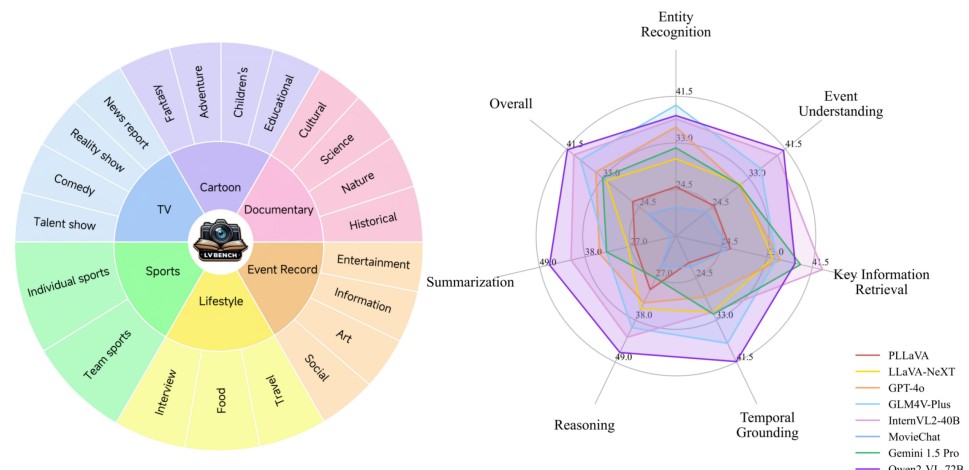

Figure 1: (Left) Video categories. Our dataset contains 6 major categories and 21 subcategories. (Right) Performance radar chart of different models on LVBench.

## Abstract

Recent progress in multimodal large language models has markedly enhanced the understanding of short videos (typically under one minute), and several evaluation datasets have emerged accordingly. However, these advancements fall short of meeting the demands of real-world applications such as embodied intelligence for long-term decision-making, in-depth movie reviews and discussions, and live sports commentary, all of which require comprehension of long videos spanning several hours. To address this gap, we introduce LVBench, a benchmark specifically designed for long video understanding. Our dataset comprises publicly sourced videos and encompasses a diverse set of tasks aimed at long video comprehension and information extraction. LVBench is designed to challenge multimodal models to demonstrate long-term memory and extended comprehension capabilities. Our extensive evaluations reveal that current multimodal models still underperform on these demanding long video understanding tasks. Through LVBench, we aim to spur the development of more advanced models capable of tackling the complexities of long video comprehension.

## 1 Introduction

Recently, the rapid advancements in large language models (OpenAI, 2023; Anthropic, 2024; Du et al., 2021) and visual feature extraction models (Radford et al., 2021; Sun et al., 2023; Zhai et al., 2023) have led to significant improvements in the performance of multimodal large models on open-domain video question-answering tasks. These multimodal understanding models have also empowered various downstream tasks, such as embodied intelligence, video generation, and subtitles for the visually impaired. However, most current end-to-end video understanding models are limited to processing videos of only a few minutes in length. More complex tasks require the capability

to understand much longer videos, which presents a significant challenge to existing multimodal models.

Despite numerous video understanding benchmarks being proposed in the past, the field of long video understanding remains underdeveloped due to the difficulties in data acquisition and annotation. To address this gap, we introduce LVBench, a benchmark designed to evaluate the capabilities of models in understanding long videos. We collected a substantial amount of long video data from public sources, with annotations provided through a combination of manual effort and model assistance. Additionally, we carefully designed a series of evaluation tasks. Compared to previous video understanding benchmarks (Li et al., 2023c), LVBench offers the following unique features:

- We define six core capabilities for long video understanding, which can be flexibly combined to create complex and challenging questions. This multifaceted approach enables a comprehensive evaluation of a model's ability to process and comprehend lengthy video content.

- We have collected a diverse range of long video data from various sources, with an average duration approximately four times longer than the longest existing dataset. The categories of videos in LVBench are illustrated in Figure 1. This extensive collection of long-form video content provides a robust foundation for testing models on extended temporal contexts.

- Through meticulous human annotation and multi-stage quality control processes, we ensure the high quality of our dataset, providing a reliable benchmark for assessing long video understanding capabilities.

## 2 RELATED WORKS

**Multi-modal Large Language Models.** Building upon the achievements in Large Language Models (LLMs), the field has shifted towards Multi-modal Large Language Models (MLLMs) to enhance multi-modal understanding and generation capabilities (Wang et al., 2023; Hong et al., 2023; Alayrac et al., 2022; Li et al., 2023b;c; Liu et al., 2024c; Xu et al., 2024). Early advancements in this area include models like Flamingo (Alayrac et al., 2022), which fused text and vision to perform exceptionally well in multimodal tasks. Subsequent models such as VideoChat (Li et al., 2023b) and VideoChatGPT (Maaz et al., 2024) began exploring the video modality, using ChatGPT (Achiam et al., 2023) to generate video instruction-tuning data for improved instruction-following capabilities. VideoChat2 (Li et al., 2023c) advanced the field by introducing a dedicated video encoder, requiring extensive training on large-scale datasets to optimize performance. The ST-LLM (Liu et al., 2024c) model streamlined this process by leveraging LLMs for visual sequence modeling, thereby reducing training complexity and enhancing performance. PLLaVA (Xu et al., 2024) introduced a resource-efficient method for adapting image-language pre-trained models to dense video understanding through a novel feature pooling strategy, achieving state-of-the-art results. Gemini 1.5 Pro (Reid et al., 2024) further pushed the boundaries with a mixture-of-experts architecture, excelling in long-context reasoning and multi-modality across extensive multimodal benchmarks. These advancements underscore the significant progress and potential of MLLMs in advancing multimodal comprehension and generation. Despite the progress made, our experiments indicate that current video understanding models still fall short on tasks requiring long-range comprehension, highlighting an urgent need for the development of models tailored for long video understanding.

**Benchmarks for MLLM.** Recent advancements in vision-language (VL) benchmarks have largely focused on images and short videos, as seen in datasets like MMBench (Liu et al., 2023), SEED-Bench-2 (Li et al., 2023a), TGIF-QA (Jang et al., 2017) and MVBench (Li et al., 2023c). For long video understanding, previous benchmarks such as Perception Test (Pătrăucean et al., 2023) have explored multi-modal video perception and reasoning but often with shorter video clips and limited temporal context. Datasets like How2QA (Li et al., 2020) and ActivityNet-QA (Yu et al., 2019) are domain-specific and do not adequately capture the complexity of long-term video understanding. EgoSchema (Mangalam et al., 2024) and MovieQA (Tapaswi et al., 2016) provide insights into narrative and thematic understanding but are constrained by shorter video durations and limited granularity. While LongVideoBench (Wu et al., 2024), MovieChat (Song et al., 2023), MoVQA (Zhang et al., 2023), and Video-MME (Fu et al., 2024) utilize longer videos to test models, their average duration is still limited to around 10 minutes. In contrast, LVBench features significantly longer

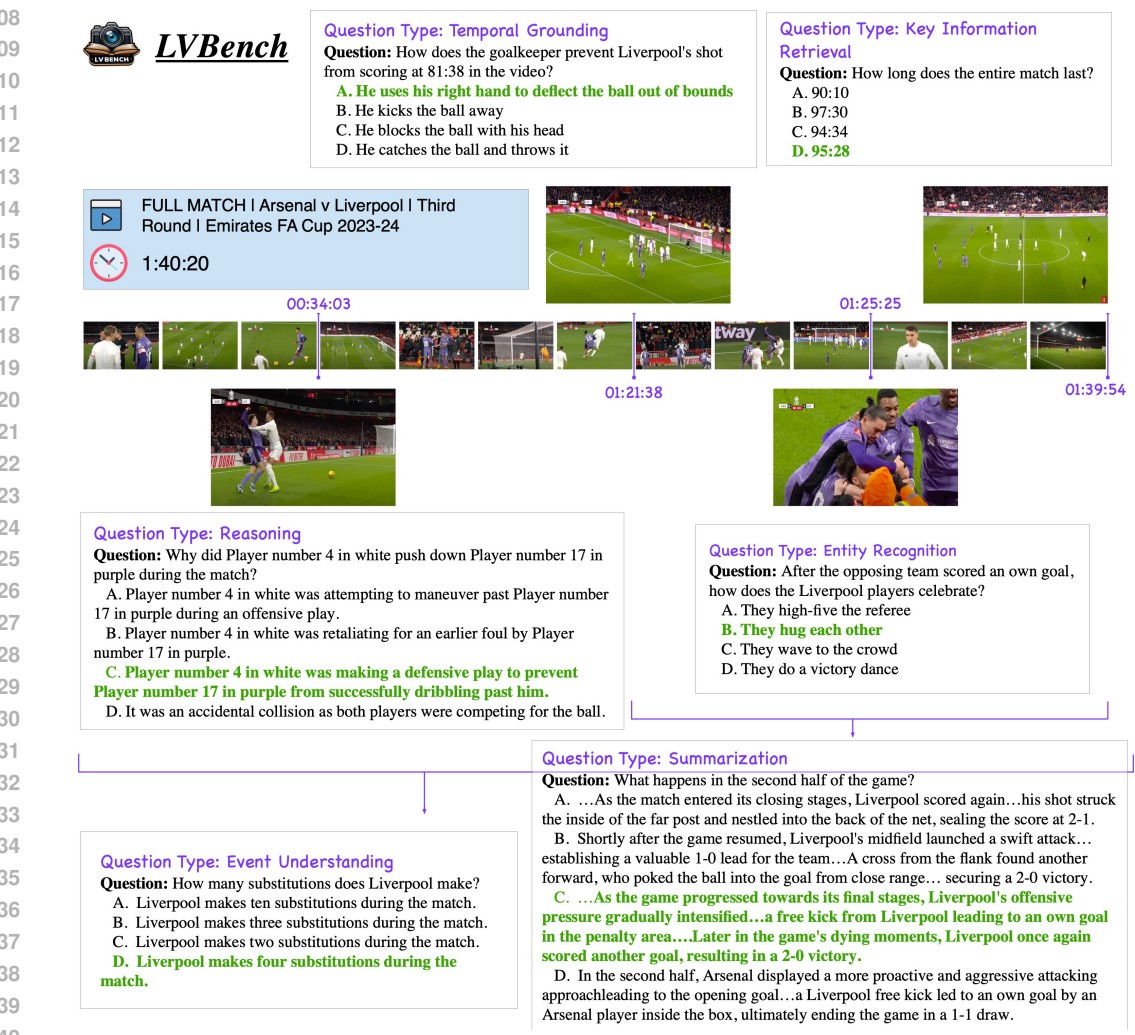

Figure 2: Examples from LVBench. LVBench covers problems involving various temporal and spatial dimensions.

video segments averaging 4101 seconds, pushing the boundaries of long-term video understanding with comprehensive tasks and detailed annotations.

# 3 LVBENCH

In this chapter, we primarily discuss the construction of the original dataset for LVBench and the generation and optimization of the question-answer pairs.

## 3.1 DATASET COLLECTION

We define a long video as having the following properties:

- A duration exceeding 30 minutes.
- Highly dynamic content with rich visual information.

To curate our dataset, we sourced publicly available videos from YouTube, covering a diverse range of topics such as sports, live streams, TV shows, documentaries, animations, and more. By using various search terms and YouTube's auto-suggestion feature, we gathered an initial collection of 500 videos, each with a minimum duration of 30 minutes. Subsequently, our annotators carefully screened these videos based on the following criteria to select a subset of 103 high-quality, diverse videos:

Table 1: Comparison of different datasets. **Open-domain** represents whether the video source is diversified. **Multi-type** represents whether the types of questions are greater than 2 categories.

| Dataset | Num QA | Avg sec | Open-domain | Multi-type | Annotation |
|---|---|---|---|---|---|
| TGIF-QA (Jang et al., 2017) | 165,165 | 3 | ✓ | ✗ | Auto |
| MSVD-QA (Xu et al., 2017) | 13,157 | 10 | ✓ | ✗ | Auto |
| MSRVTT-QA (Xu et al., 2017) | 72,821 | 15 | ✓ | ✗ | Auto |
| MVBench (Li et al., 2023c) | 4,000 | 16 | ✓ | ✓ | Auto |
| Perception Test (Pătrăucean et al., 2023) | 44,000 | 23 | ✗ | ✓ | Auto&Manual |
| NExT-QA (Xiao et al., 2021) | 52,044 | 44 | ✓ | ✗ | Manual |
| How2QA (Li et al., 2020) | 44,007 | 60 | ✓ | ✓ | Manual |
| ActivityNet-QA (Yu et al., 2019) | 800 | 111 | ✗ | ✗ | Manual |
| CinePile (Rawal et al., 2024) | **303,828** | 160 | ✗ | ✓ | Auto&Manual |
| EgoSchema (Mangalam et al., 2024) | 5,000 | 180 | ✗ | ✓ | Auto&Manual |
| MovieQA (Tapaswi et al., 2016) | 6,462 | 203 | ✗ | ✓ | Manual |
| LongVideoBench (Wu et al., 2024) | 6,678 | 473 | ✓ | ✓ | Manual |
| MovieChat-1K (Song et al., 2023) | 13,000 | 564 | ✗ | ✓ | Manual |
| MoVQA (Zhang et al., 2023) | 21,953 | 992 | ✗ | ✓ | Manual |
| Video-MME (Fu et al., 2024) | 2,700 | 1018 | ✓ | ✓ | Manual |
| **LVBench(Ours)** | 1,549 | **4,101** | ✓ | ✓ | Manual |

- The presence of one or more protagonists (possibly in a first-person perspective) who serve as narrators, appearing on-screen for a significant portion of the video and interacting with the environment.

- A complete video structure with a coherent logical flow.

- The occurrence of multiple minor events throughout the video, following a chronological order and exhibiting completeness.

- Visuals that are relatively easy to comprehend without overly fragmented information.

- Video content that can be understood independently of audio cues.

This multi-stage filtering process ensures that our dataset comprises diverse, high-quality long videos suitable for evaluating complex video understanding tasks.

## 3.2 TASK DEFINITION

To comprehensively evaluate long video understanding, we propose a benchmark testing six core capabilities. Example questions for each capability are presented in Figure 2. The proportion of each capability is shown in Figure 3. Questions are designed to flexibly combine multiple skills to construct complex queries that probe a model's capacity to:

1. **Temporal Grounding(TG)**: Questions focus on understanding sequences and dynamics within the video, such as identifying specific events at designated times (e.g., *"What happened at 29:30?"*).

2. **Summarization(Sum)**: Annotators are required to produce an abstractive summary that encapsulates the entire video content, demonstrating a cohesive understanding of the sequence from start to finish.

3. **Reasoning(Rea)**: This involves the application of advanced reasoning skills to interpret the video content:
   - **Causal**: Determining causal relationships between events.
   - **Emotional**: Understanding the emotional developments of characters.
   - **Intentional**: Interpreting the underlying intentions of characters.
   - **Prospective**: Making predictions about future events based on current evidence.

4. **Entity Recognition(ER)**: This capability requires the identification and continuous tracking of key entities (such as people, places, and objects) throughout the video:
   - **Entity Detection**: Identifying mentions of entities and resolving their identities across different instances.

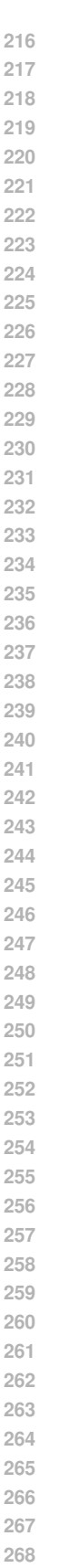

Figure 3: The proportion of different core capabilities.

- **Relation Extraction**: Extracting the relationships among identified entities.
- **Action Recognition**: Observing and understanding the progression of an entity's actions over time.
- **Entity Association**: Linking entities to relevant events.

5. **Event Understanding(EU)**: Comprehending overarching semantic concepts in the video:

- **Video Classification**: Classifying the genre of the video (e.g., news, film).
- **Event Detection**: Identifying significant occurrences within the video (such as goals scored or conflicts).
- **Scene Perception**: Recognizing changes between different scenes or settings.

6. **Key Information Retrieval(KIR)**: Extracting specific, detailed information, such as text displayed in the video (e.g., *"What revenue growth did the firm report at the conference?"*).

By composing questions that test combinations of these temporal, abstractive, reasoning, entity-centric, event-based, and detail-oriented skills, our benchmark enables robust evaluation of a model's ability to understand long videos across multiple modalities. This multifaceted typology covers the key cognitive capabilities required to comprehend complex, open-ended video.

## 3.3 QA GENERATION

The total number of questions for each video is positively correlated with the video duration, averaging 24 questions per hour. After constructing a question, annotators are asked to provide four options, including one correct answer and three distractors. Annotating long videos is significantly more challenging than annotating short videos or image data, making quality control a substantial challenge. To ensure the quality of the evaluation set, we encourage annotators to follow these principles:

- **Question Diversity**: Construct at least one question for each question type in a video.
- **Specificity**: Questions should refer to unique scenes, events, or characters, avoiding vague descriptions. For example, if a video contains two arguments, a well-constructed question might be: *"When did A and B start arguing?"* or *"How did the person in red's expression change during the hallway argument?"*. Less specific questions would be: *"Why did they start arguing?"* or *"Who are the people arguing?"*
- **Temporal Coverage**: Questions should cover multiple events throughout the video, avoiding repetition of a single event.
- **Consistency**: Constructed correct answers should precisely address the questions. Answers should match the content of the questions, avoiding irrelevant information. Correct and incorrect answers should be constructed consistently, avoiding obvious differences in length, form, or format.

By following these principles, we aim to create a high-quality, diverse, and challenging evaluation set for long video understanding.

### 3.4 DATA QUALITY CONTROL

During the annotation process, we observed that annotators had a tendency to label most questions as temporal grounding, i.e., specifying a time range to limit the referent of the question. This practice may inadvertently reduce the difficulty of the questions and unfairly disadvantage video-understanding models that lack the ability to perceive the temporal dimension. To address this issue, we instructed annotators to minimize the number of temporal questions while still ensuring the uniqueness of the referents, effectively converting temporal grounding questions into other question types.

Upon constructing all the questions, we discovered that for certain questions, a language model could generate answers without any visual input. As highlighted in MMstar (Chen et al., 2024a), many multimodal benchmarks can be effectively solved using pure text input alone. To mitigate this issue, we employed a straightforward yet effective approach. We utilized two powerful large language models, GLM-4 (Du et al., 2021) and GPT-4 (Achiam et al., 2023), to independently generate answers for all the questions. In cases where the outputs from both models were identical and matched the ground truth answer, we removed that particular data sample from the dataset. This filtering process successfully eliminated the majority of questions that did not rely on video content for answering. Following this filtering step, we obtained a refined set of 1,549 question-answer pairs.

## 4 EXPERIMENTS

In this chapter, we report the experimental results of different video understanding models on LVBench and also compare them with human performance.

### 4.1 SETTINGS

We evaluated the performance of nine models that support multi images or short video input: TimeChat (Ren et al., 2023), Video-ChatGPT (Maaz et al., 2023), PLLaVA (Xu et al., 2024), LLaVA-Onevision (Li et al., 2024), CogVLM2-Video (Hong et al., 2024), LLaVA-NeXT (Zhang et al., 2024), GPT-4o (OpenAI, 2024), GLM4V-Plus (Hong et al., 2024) and InternVL2-40B (Chen et al., 2024b). To adapt these models for long video inputs, we sample a fixed number of frames from the original video, such as 32 or 96 frames, to maintain consistency with the model's training sequence length. Additionally, we assessed six models that natively support long videos: LLaMA-VID (Li et al., 2023d), MovieChat (Song et al., 2023), LWM (Liu et al., 2024a), Gemini 1.5 Pro (Reid et al., 2024), Kangaroo (Liu et al., 2024b) and Qwen2-VL-72B (Wang et al., 2024). We processed the videos at a rate of 1 frame per second and fed them into the models, only performing downsampling when the video's length exceeded the model's maximum processing capability. It is worth noting that although Gemini 1.5 Pro can handle videos up to 10 hours long, its publicly available interface is limited to processing videos of up to 1 hour in length. For each question, we provided the following prompt as input to the models:

> *Question (A) Option1 (B) Option2 (C) Option3 (D) Option4. Please select the best answer from the options above and directly provide the letter representing your choice without giving any explanation.*

After obtaining the model responses, we first attempted to extract the answers using regular expression matching. For questions where the matching process was unsuccessful, we employed a GLM-4 model to extract the answers from the responses.

### 4.2 EVALUATION RESULTS

#### 4.2.1 PERFORMANCE ACROSS CORE CAPABILITIES

To comprehensively evaluate the performance of various long video understanding models across core capabilities, we conducted extensive experiments on the LVBench dataset, testing multiple repre-

Table 2: LVBench evaluation results regarding each core long video understanding capability. The highest score are highlighted with green, and the second highest are highlighted with purple. All the numbers are presented in % and the full score is 100%.

| Model | LLM | ER | EU | KIR | TG | Rea | Sum | Overall |
|---|---|---|---|---|---|---|---|---|
| *Non-Native Long Video Support Models* | | | | | | | | |
| TimeChat (Ren et al., 2023) | LLaMA2-7B | 21.9 | 21.7 | 25.9 | 22.7 | 25.0 | 24.1 | 22.3 |
| Video-ChatGPT (Maaz et al., 2023) | Vicuna-1.5-13B | 22.9 | 22.6 | 22.7 | 25.5 | 23.4 | 24.1 | 23.1 |
| PLLaVA (Xu et al., 2024) | Yi-34B | 25.0 | 24.9 | 26.2 | 21.4 | 30.0 | 25.9 | 26.1 |
| LLaVA-OneVision (Li et al., 2024) | LLaMA3-70B | 25.0 | 26.9 | 29.2 | 30.9 | 25.4 | 31.0 | 26.9 |
| CogVLM2-Video (Hong et al., 2024) | LLaMA3-8B | 28.3 | 26.9 | 31.0 | 25.1 | 25.5 | 38.9 | 28.1 |
| LLaVA-NeXT (Zhang et al., 2024) | Yi-34B | 30.1 | 31.2 | 34.1 | 31.4 | 35.0 | 27.6 | 32.2 |
| GPT-4o (OpenAI, 2024) | GPT-4 | 35.9 | 30.8 | 35.5 | 28.3 | 33.5 | 34.5 | 34.7 |
| GLM4V-Plus (Hong et al., 2024) | GLM-4 | 39.9 | 35.8 | 34.8 | 37.7 | 40.0 | 32.8 | 38.3 |
| InternVL2-40B (Chen et al., 2024b) | Nous-Hermes-2-Yi-34B | 37.4 | 39.7 | 43.4 | 31.4 | 42.5 | 41.4 | 39.8 |
| *Native Long Video Support Models* | | | | | | | | |
| MovieChat (Song et al., 2023) | Vicuna-7B | 21.3 | 23.1 | 25.9 | 22.3 | 24.0 | 17.2 | 22.5 |
| LLaMA-VID (Li et al., 2023d) | Vicuna-13B | 25.4 | 21.7 | 23.4 | 26.4 | 26.5 | 17.2 | 23.9 |
| LWM (Liu et al., 2024a) | LLaMA2-7B | 24.7 | 24.8 | 26.5 | 28.6 | 30.5 | 22.4 | 25.5 |
| Gemini 1.5 Pro (Reid et al., 2024) | Gemini 1.5 Pro | 32.1 | 30.9 | 39.3 | 31.8 | 27.0 | 32.8 | 33.1 |
| Kangaroo (Liu et al., 2024b) | LLaMA3-8B | 38.6 | 37.9 | 29.6 | 35.0 | 41.3 | 36.2 | 38.3 |
| Qwen2-VL-72B (Wang et al., 2024) | Qwen2-72B | 38.0 | 41.1 | 38.3 | 41.4 | 46.5 | 46.6 | 41.3 |

sentative models, including both non-native and native long video support models. The experimental results are presented in Table 2.

Overall, Qwen2-VL-72B achieved the best performance, outperforming other models in multiple tasks such as entity understanding (EU), temporal grounding (TG), reasoning (Rea) and summarization (Sum). Interestingly, some models that do not natively support long videos still managed to achieve competitive results compared to native long video support models. In terms of the overall score, InternVL2-40B ranked second only to Qwen2-VL-72B, and GLM4V-Plus ranked third, tying with the native long video support model Kangaroo.

However, the results of three widely used long video models, LLaMA-VID, MovieChat, and LWM, were nearly equivalent to random selection, highlighting the significant challenges that current models face when processing extremely long videos. This suggests that despite the ability to input long videos through model structure optimization, the performance and effectiveness of these models have not substantially improved.

### 4.2.2 ANSWER DISTRIBUTION

To understand why some native long video support models perform poorly on LVBench, we evaluated the distribution of answers generated by different models on LVBench and observed that existing long video understanding models struggle with precisely following instructions. For example, despite explicitly constraining the output in the prompt to be one of four provided answer choices, Gemini 1.5 Pro generated responses outside of the specified options 20.9% of the time, such as *"None of the above*

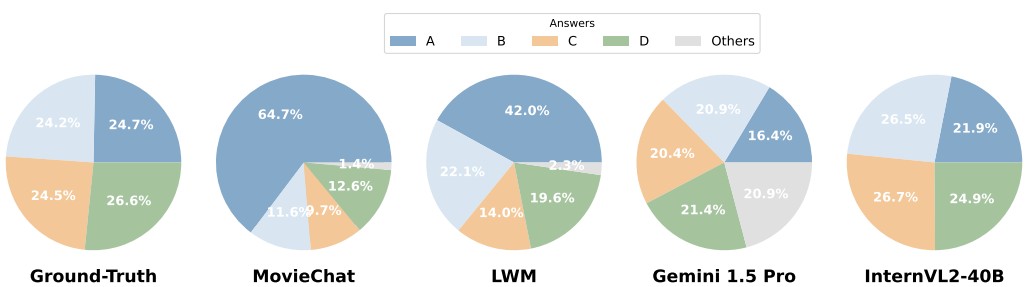

Figure 4: Distribution of answers generated by different models.

*options are correct"* or *"I cannot answer this question".* This occurred even though manual validation confirmed that the questions were indeed answerable from the given choices. MovieChat and LWM exhibited a strong bias towards selecting option A, regardless of the question. In contrast, InternVL2-40B demonstrated the strongest instruction-following capability, never generating responses outside the constrained options and producing a nearly uniform distribution over the answer choices.

We hypothesize that this discrepancy arises from the relatively higher quality and diversity of image-based instruction fine-tuning data compared to video instruction data. As InternVL2-40B ingests fewer image inputs, it can more readily generalize the learned capability of precisely following instructions from the image modality to video.

## 4.3 PERFORMANCE ACROSS VIDEO CATEGORIES

We conducted a comprehensive evaluation across various video categories. We selected two state-of-the-art models, InternVL2-40B and Qwen2-VL-72B, for testing and compared their results with human performance.

As shown in Table 3, humans achieved very high accuracy across all video categories, with an average of 94.4%. In contrast, the overall performance of InternVL2-40B and Qwen2-VL-72B was relatively lower, at only 39.5% and 41.3%, respectively. This indicates that there is still a significant performance gap between current multimodal models and humans in video understanding tasks, suggesting considerable room for improvement in understanding and analyzing long video content.

Table 3: LVBench evaluation results across different video categories.

| Model | Sports | Documentary | Event Record | Lifestyle | TV Show | Cartoon | Overall |
|---|---|---|---|---|---|---|---|
| Human | 96.3 | 89.8 | 87.4 | 98.4 | 97.2 | 95.8 | 94.4 |
| InternVL2-40B | 43.5 | 45.2 | 38.9 | 41.6 | 32.8 | 36.4 | 39.5 |
| Qwen2-VL-72B | 43.0 | 42.6 | 40.8 | 41.0 | 42.0 | 38.9 | 41.3 |

Further analysis of the results for each category revealed that InternVL2-40B performed best on documentary videos, reaching an accuracy of 45.2%, while performing worst on TV shows, with only 32.8%. Qwen2-VL-72B, on the other hand, achieved the highest accuracy of 43.0% on sports videos and the lowest performance of 38.9% on cartoon.

## 4.4 IMPACT OF LLM FILTERING METHOD

Table 4 demonstrates the effectiveness of using large language models to filter question-answer pairs. Despite instructing the annotators to watch the entire video before labeling, a significant proportion of questions could still be answered correctly by inferring from the matching degree between the question and options, as

Table 4: Ablation study on LLM filtering method.

| Model | w/ LLM | w/o LLM |
|---|---|---|
| LWM | 25.5 | 32.7 |
| LLaVA-NeXT | 32.2 | 48.9 |

well as the differences among the options. The experimental results show that after applying the LLM filter, the score of LWM decreased from 32.7% to 25.5%, while the score of LLaVA-NeXT, which employs a more powerful language model, dropped from 48.9% to 32.2%, with a decline of 16.7 percentage points. This indicates that stronger language models have a higher probability of inferring the correct answer solely from the natural language context, highlighting the importance of the data filtering step in the process.

## 4.5 IMPACT OF VIDEO AND CLUE LENGTH

We investigated the impact of different video durations and clue durations on the experimental results. As shown in Figure 5, the performance of GLM4V-Plus, InternVL2-40B, and Qwen2-VL-72B remains relatively stable across various video length intervals, demonstrating overall strong performance.

Clue durations refers to the time span of video content needed to answer a specific question. Figure 5 illustrates that most models perform well when the clue duration is between 0-10 seconds or greater than 60 seconds. This may be attributed to the fact that questions with clues longer than 60 seconds tend to focus more on analyzing and summarizing the relationships between multiple events. These models are equipped with stronger language modeling capabilities, giving them an advantage in tackling such problems.

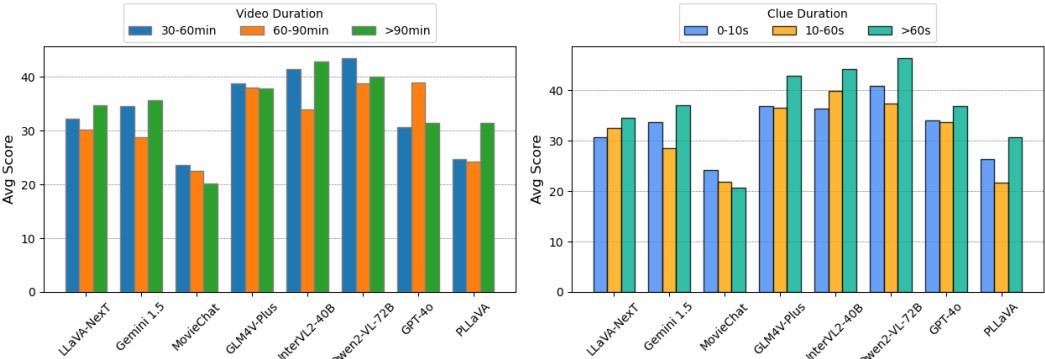

Figure 5: The impact of different video and clue durations.

## 5 DISCUSSION

**Conclusion.** In this paper, we introduced LVBench, a benchmark designed to advance long video understanding. LVBench comprises a diverse collection of lengthy videos and a meticulously annotated question-answer dataset, presenting a robust evaluation framework for assessing multimodal models on complex video understanding tasks. Our experiments revealed that while state-of-the-art models have made strides in short video understanding, their performance on long videos falls short of human-level accuracy. By providing a challenging benchmark, we hope to stimulate the development of advanced models capable of tackling the complexities of extended video comprehension.

**Limitations.** A limitation of our benchmark is the exclusion of audio data. While audio can provide valuable context, we did not include it because most current models lack effective audio processing capabilities. Future work will aim to incorporate audio information to enhance the evaluation framework for multimodal understanding.

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

# A APPENDIX

## A.1 DISTRIBUTION OF CORE CAPABILITY COMBINATIONS

In this section, we quantified the distribution of various combinations of core competencies within the dataset. As illustrated in the Figure 6, the six core competencies can be further combined to form 26 fine-grained question types. This flexible combinatorial approach guarantees the richness and diversity of the dataset, enabling a comprehensive evaluation of the model's performance across multiple dimensions.

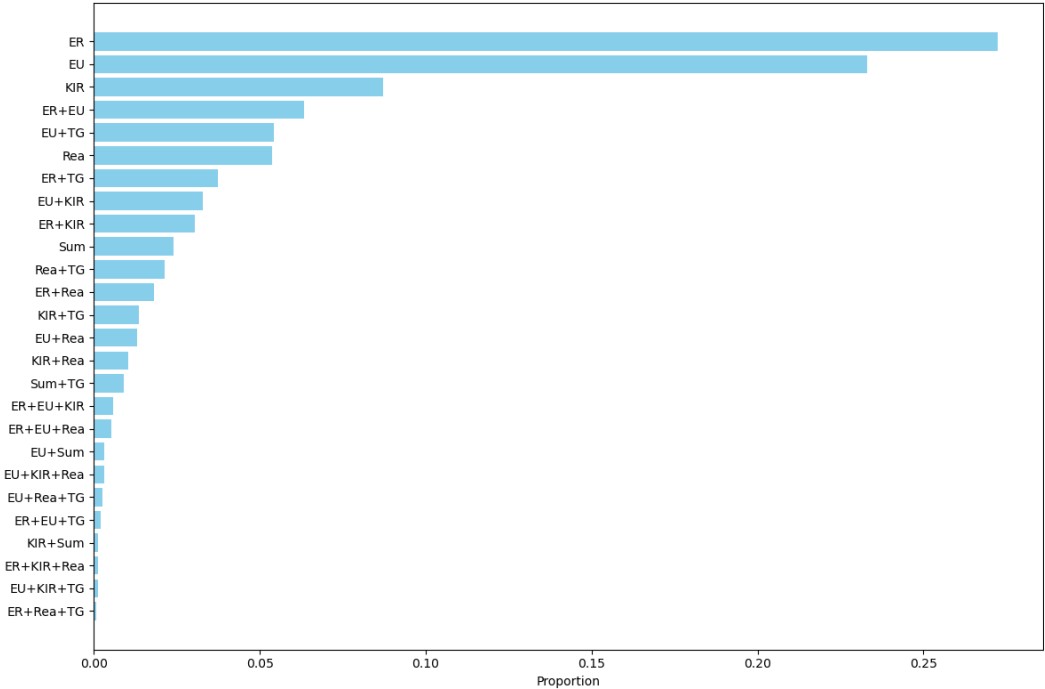

Figure 6: The proportion of core capability combinations.

