# OpenReview forum: "LVBench: An Extreme Long Video Understanding Benchmark"
_ICLR.cc/2025/Conference — ICLR 2025 Conference Withdrawn Submission_

### Official Review · Reviewer_qFeo · 2024-11-02

**Soundness:** 3
**Presentation:** 3
**Contribution:** 3
**Rating:** 5
**Confidence:** 4

**Summary:**

Authors propose a long-video QnA evaluation benchmark consisting of human annotated video question answer pairs. They evaluate multiple state-of-the-art approaches on this dataset, highlighting the difficulty of the benchmark for existing approaches. The authors clearly highlight the distinctions of this benchmark compared to prior work, focussing on video length, diversity, and question complexity. Several interesting analysis is provided on the dataset.

**Strengths:**

1. Timely and useful benchmark for long-video QnA containing diverse and long videos
2. Paper is quite well written clearly outlining the reasoning for building this benchmark and how it differentiates from existing works. Table 1 in particular is quite useful for the latter.
3. Several interesting analysis on dataset statistics.

**Weaknesses:**

1. L143: Consider adding hours, i.e.  … 4101 seconds, … -> … 4101 seconds ( > one hour), …

2. **On L316 (answer matching to choices):** The authors use the following setup for selecting the correct choice with a given VLM: “After obtaining the model responses, we first attempted to extract the answers using regular expression matching. For questions where the matching process was unsuccessful, we employed a GLM-4 model to extract the answers from the responses.”
    1. Could you try likelihood selection / cloze-prompting similar to most NLP benchmarks (see likelihood selection in [1] or cloze-prompting in [2])? This could be more efficient to evaluate (as opposed to generating tokens iteratively) and also avoid any negatives due to difficulty in answer matching.
    2. Could you explain how the GLM-4 model is used exactly to extract answers? Are you directly prompting GLM-4 to select one answer given the outputs of the model?
    3. If yes (to 2 above), could you also try using some text-encoder feature-space similarity between generated answer and choices (i.e. standard zero-shot setup in CLIP)?

3. **Naive Baselines:** Can we include two naive baselines with only text and a single center frame? These may give some interesting insights about the benchmark.
    1. For only text, evaluate with a SOTA LLM (feeding no images, using only the question and choices) similar to what is done in [1, 3]. See Just LLM in [1] and blind variant in Table 4 of [3].
    2. For single center frame, using some SOTA VLM and feed only the single center frame (this is motivated by Single Frame VLM in [1]).

4. Table 2: Could you repeat the abbreviation names (ER EU KIR TG Rea Sum) in the caption? This would make it easier for a reader.

5. Is performance on the dataset actually correlated to long video understanding ability or instruction following ability of evaluated models?
    1. This is motivated by L379-382 mentioning “MovieChat and LWM exhibited a strong bias towards selecting option A, regardless of the question. In contrast, InternVL240B demonstrated the strongest instruction-following capability, never generating responses outside the constrained options and producing a nearly uniform distribution over the answer choices.”
    2. Maybe a secondary set of results using likelihood selection (see point 2.1 above) may resolve this problem. Integrating such an evaluation scheme into the benchmark could help it better stand out from existing long-video benchmarks making evaluation both faster and more robust.

6. Can you explain LLM filtering (Section 4.4) in more detail? If this is related to point 3, adding a full row of that to Table 2 could provide a lot more insights, strengthening the benchmark.

7. “Clue Duration” : can you explain how this is calculated? Is this part of the dataset ground-truth provided by humans?

8. Similar to Figure 2, maybe consider adding visual examples for each question type in the appendix.

9. Also consider adding visual examples for each of the 6 major categories in the appendix. And possibly link to examples of all 21 sub-categories also in some external website.

10. L072 mentions “Through meticulous human annotation and multi-stage quality control processes …” for dataset curation. Please explain this process in more detail.

11. Compute, Inference Time, and other details
    1. For a reader, it will be useful to know the compute required to evaluate a model on this benchmark. Consider elaborating on compute used for evaluation and time taken.
    2. Details on dataset size (i.e. storage required) and also any copyrights details for any videos used will be useful for those using benchmark in future.

&nbsp;

[1] “Understanding Long Videos in One Multimodal Language Model Pass.” ArXiv abs/2403.16998

[2] “Leveraging Large Language Models for Multiple Choice Question Answering.” ICLR 2023

[3] “Memory Consolidation Enables Long-Context Video Understanding.” ArXiv abs/2402.05861

**Questions:**

1. The additional explanations regarding various sections under weaknesses
2. How would alternate evaluation schemes like likelihood based selection affect performance? This ties into Q5 above: is performance on the dataset actually correlated to long video understanding ability or instruction following ability of evaluated models?
3. Possibly related to LLM filtering idea; how would the naive baselines (suggested under weaknesses) perform on the benchmark?

This proposed dataset / benchmark would definitely be valuable for the community to better evaluate long video understanding approaches. It would be really great if the authors could clarify / address the above points.

---

> ### Author Response · Authors · 2024-11-15
> **Response to Reviewer qFeo**
>
> We sincerely appreciate your constructive feedback and insightful comments. Your suggestions will greatly help us improve the quality of our paper. We will carefully address all the points raised and revise the paper accordingly.

---

### Official Review · Reviewer_cRZL · 2024-11-03

**Soundness:** 3
**Presentation:** 2
**Contribution:** 3
**Rating:** 5
**Confidence:** 4

**Summary:**

This is a benchmark dataset paper. The paper provides multiple choice question annotations of 30+ minute videos, designing them to be only used for the evaluation. The videos in the dataset are certainly longer than previous datasets.

**Strengths:**

- It is a new benchmark with longer videos than other existing datasets.

**Weaknesses:**

- The annotation process is not 100% clear. The paper does not reveal the detailed process used to collect the annotations. For instance, what instructions were given to the annotators to prepare the question and the four options in multiple choice questions? Example annotations provided in the paper are also extremely limited (just one example in Figure 2 per type), making it very difficult to judge the quality of the dataset. We also do not see any supplementary material or appendix with such examples.

- It seems the dataset only has multiple choice questions.

- There is no video license consideration with the videos in the dataset. In fact, the paper does not mention licensing at all. How do the authors plan to distribute the videos in this dataset? The YouTube videos get deleted occasionally, and the benchmark accuracy will be influenced by what videos are still remaining on YouTube. This happened in previous datasets like Kinectic400 before, and this is concerning. In addition, this will restrict people from using this dataset for any model training.

- Since the paper mentions movies as some of its videos, please include other relevant datasets like Short Film Dataset (SFD).

**Questions:**

Please discuss the video license issues.

**Details Of Ethics Concerns:**

The paper does not discuss about the copyright and licensing of the videos used to construct this dataset at all. How do they plan to distribute the dataset without clearing it? This has to be resolved.

---

> ### Author Response · Authors · 2024-11-15
> **Response to Reviewer cRZL**
>
> We sincerely appreciate your constructive feedback and insightful comments. Your suggestions will greatly help us improve the quality of our paper. We will carefully address all the points raised and revise the paper accordingly.

---

### Official Review · Reviewer_n5MG · 2024-11-04

**Soundness:** 3
**Presentation:** 4
**Contribution:** 2
**Rating:** 3
**Confidence:** 5

**Summary:**

The paper proposes a long video understanding benchmark for multimodal large language models (MLLMs), focussing on videos over 30 minutes. The benchmark includes ~100 diverse videos with ~1550 QA paiers that test six core capabilities: temporal grounding, summarization, reasoning, entity recognition, event understanding, and key information retrieval. The paper studies several state-of-the-art MLLMs, both that focus on long videos natively and ones that don't. They key focus is on understanding the capabilities (or lack thereof) of current models in understanding lengthy video content.

**Strengths:**

+ The paper introduces a video understanding benchmark that emphasizes long videos, longer than prior works.
+ The paper does a commendable job of enumerating different capabilities required for long video understanding
+ The paper evaluates a total of 15 different MLLMs, which is a reasonably comprehensive list.

**Weaknesses:**

Benchmark:
- While I appreciate the effort to create a benchmark with longer videos, as emphasized in the paper and Table 1, without enough scale, the benchmark can prove to be of limited utility. As noted in Table 1, the benchmark as 1549 QA pairs, making it the second smallest benchmark in terms of number of QA, with only the older ActivityNet-QA (Yu et al., 2019) having fewer QA pairs. It only has 103 videos, which is considered quite small by current standards. In fact, number of videos should have been a column in Table 1 to compare with existing  benchmarks, which would reveal how small the dataset is. While the videos in the proposed benchmark are longer, a large number of videos is still necessary for a credible video understanding benchmark, especially since the QA pairs in this paper are video-level rather than fixed-time-clip-level.
- Although the proposed benchmark includes 21 video subcategories, it still lacks many common and significant types of videos, such as video games, non-cartoon educational videos, travel videos/vlogs, health and fitness videos, and technology videos, etc. Not necessarily a negative, but the paper should discuss other potential categories it does not cover so folks who use the results on benchmark should understand what it works or doesn't work on.
- On a similar note, the benchmark is heavily skewed towards object and action recognition -- arguably two of the most studied tasks in recognition (Figure 3, >80% of questions come from Entity and Event recognition categories).
- Another baseline missing is the following. Using video (or clip) caption generation methods to describe the video and then use LLMs to answer the questions using those descriptions/captions as input.

Exposition
- The paper emphasizes several times that the questions are designed to be compositional, which can flexibly combine multiple skills to construct complex queries. However, from what I gather, all the questions belong to one of the categories listed in Section 3.2, and there's no composition of questions. It might be a language issue but compositionality of questions has a specific meaning and I'd recommend paraphrasing to not imply compositionality.
- I like the "Clue duration" analysis, however, I don't understand how they ground-truth clue duration was collected? Were annotators asked to label this or was this estimated in an automated way?

Experimentation
- The paper discusses some details in L302-4 regarding how they adapt models not designed long videos. Example, by sampling 32 or 96 frames to provide as input. Now, I understand that if the models are not designed for long videos and/or have limited context length, then it's not easy to adapt MLLMs for long videos. However, providing 32 or 96 frames for avg. ~4000 seconds, without any sampling of frames, is a futile exercise when it comes to understanding long videos. The paper should either propose ways in which to adapt models that natively do not support long videos that is more meaningful or not have so many of them and focus on models that do support longer videos.
- I like the analysis in Section 4.2.2, but I it is still unclear how much of the answer/distribution comes from actually understanding videos vs. LLMs inherent biases. It would help to use the LLMs for each model to answer the questions and provide that as lower bound for the results. I understand the authors did try filtering out questions using two LLM models, but for this to make sense, it has to be done using the LLM underlying the MLLM.
- InternVL2-40B results: Why's there a discrepancy between Table 2 (39.8) vs. Table 3 (39.5).
- Table 3 -- why can't these numbers be provided for all method studied as opposed to just the two best methods? It would help understand the video category axes of performance for all methods, and doesn't require any more experimentation.

Findings:
- While a pure benchmark and result paper is suffice, it would be good if the paper provided some discussion on potential directions to improve current MLLMs using the proposed benchmark. Another shortcoming is straight foward application of MLLMs not designed for long video understanding. I'd have preferred to see some effort to make these MLLMs work better on long video understanding.

Minor typos:
Section 3.2, all enumerated items are missing a space before the parenthesis (Grounding(TG) --> Grounding (TG)).

**Questions:**

Please see the weaknesses listed.

Minor questions:
- Clarify how the "Clue duration" ground-truth was collected.
- InternVL2-40B results: Why's there a discrepancy between Table 2 (39.8) vs. Table 3 (39.5).
- Table 3 -- why can't these numbers be provided for all method studied as opposed to just the two best methods?

---

> ### Author Response · Authors · 2024-11-15
> **Response to Reviewer n5MG**
>
> We sincerely appreciate your constructive feedback and insightful comments. Your suggestions will greatly help us improve the quality of our paper. We will carefully address all the points raised and revise the paper accordingly.

---

### Official Review · Reviewer_8kpQ · 2024-11-04

**Soundness:** 2
**Presentation:** 2
**Contribution:** 2
**Rating:** 5
**Confidence:** 4

**Summary:**

In this paper, the authors introduce LVBench, a new MLLM benchmark designed to evaluate performance on long-term videos. LVBench comprises 103 high-quality videos sourced from YouTube, each with an average duration exceeding one hour. These videos are manually annotated using a multiple-choice question-answer format. The question types are diverse, including grounding, key information retrieval, and complex reasoning across events. In the experimental section, several representative MLLMs that support both short and long videos are evaluated and compared using the LVBench benchmark.

**Strengths:**

* Serveral baseline methods are adopted and tested on the new benchmark.
* The quality control process is well designed to avoid the imbalance of question types and language bias.

**Weaknesses:**

I have serveral concerns regarding the quality of the benchmark and insufficent exploration of the bottleneck of current video MLLMs.

* Diversity of Videos: Although the benchmark includes 103 extensively long videos from YouTube, concerns remain regarding the diversity of scene categories, filming techniques, event relationships, and the variety of objects and motions depicted. adding a statistical chart to illustrate the diversity of the videos is necessary.

* Double checking. The dataset would benefit from manual cross-checking to identify any annotation errors raised by a single annotator.

* Few-Frame Bias: A key feature of this benchmark is the extended duration of videos, which leads to complex events and dense question-and-answer pairs. Unlike short-term video benchmarks that can often be addressed with a few frames (resulting in a single-frame bias), long-term benchmarks contain more extensive information. An analysis of the number of input frames and the impact of "few-frame bias" would provide insights.

* Ambiguity in Questions: I think the utimate goal of such benchmarks is to assess real-world interactions between humans and AI when engaging with long videos. However, human questions are usually ambiguous, such as “Why did they start arguing?” rather than more detailed questions like “How did the person in red’s expression change during the hallway argument?” Incorporating the ability to handle such references could be more practical.

* Missing important details. The details of the calculation of human scores in Table 3 is not given. How to determine the clue length in the video given a free-form question? Details are not given.

**Questions:**

* In Figure 5, MovieChat, which is designed to tackle long videos, performs better on short videos than long ones. The results appears counterintuitive.

* In L184, terms such as “complete video structure” and “coherent logical flow” are not clearly defined

---

> ### Author Response · Authors · 2024-11-15
> **Response to Reviewer 8kpQ**
>
> We sincerely appreciate your constructive feedback and insightful comments. Your suggestions will greatly help us improve the quality of our paper. We will carefully address all the points raised and revise the paper accordingly.

---

### Note · Authors · 2024-11-15

I have read and agree with the venue's withdrawal policy on behalf of myself and my co-authors.